# Cortical and Trabecular Bone Modeling and Implications for Bone Functional Adaptation in the Mammalian Tibia

**DOI:** 10.3390/bioengineering11050514

**Published:** 2024-05-20

**Authors:** Meir M. Barak

**Affiliations:** Department of Veterinary Biomedical Sciences, College of Veterinary Medicine, Long Island University, Brookville, NY 11548, USA; meir.barak@liu.edu

**Keywords:** bone modeling, bone functional adaptation, trabecular bone, cortical bone, tibia

## Abstract

Bone modeling involves the addition of bone material through osteoblast-mediated deposition or the removal of bone material via osteoclast-mediated resorption in response to perceived changes in loads by osteocytes. This process is characterized by the independent occurrence of deposition and resorption, which can take place simultaneously at different locations within the bone due to variations in stress levels across its different regions. The principle of bone functional adaptation states that cortical and trabecular bone tissues will respond to mechanical stimuli by adjusting (i.e., bone modeling) their morphology and architecture to mechanically improve their mechanical function in line with the habitual in vivo loading direction. This principle is relevant to various research areas, such as the development of improved orthopedic implants, preventative medicine for osteopenic elderly patients, and the investigation of locomotion behavior in extinct species. In the present review, the mammalian tibia is used as an example to explore cortical and trabecular bone modeling and to examine its implications for the functional adaptation of bones. Following a short introduction and an exposition on characteristics of mechanical stimuli that influence bone modeling, a detailed critical appraisal of the literature on cortical and trabecular bone modeling and bone functional adaptation is given. By synthesizing key findings from studies involving small mammals (rodents), large mammals, and humans, it is shown that examining both cortical and trabecular bone structures is essential for understanding bone functional adaptation. A combined approach can provide a more comprehensive understanding of this significant physiological phenomenon, as each structure contributes uniquely to the phenomenon.

## 1. Introduction

Bone tissue is an active living tissue that continuously responds to internal and external stimuli through bone modeling and bone remodeling [1,2]. Bone modeling involves the addition of bone material through osteoblast-mediated deposition or the removal of bone material via osteoclast-mediated resorption in response to perceived changes in external mechanical stimuli (i.e., mechanical loads) by osteocytes. This process is characterized by the independent occurrence of deposition and resorption, which can take place simultaneously at different locations within the bone due to variations in stress levels across its different regions. The process of bone structural adjustment and modeling is referred to as the principle of “bone functional adaptation”. This principle predicts that bone responds to repeated dynamic mechanical stimuli or the lack of them (i.e., disuse, such as in lengthy bed rest, prolonged free falling, or a microgravity environment (astronauts) [3]), by adjusting its external shape (i.e., cortical bone modeling), its internal architecture (i.e., trabecular bone modeling), its tissue structure (i.e., cortical bone remodeling), and its material properties (mainly, mineralization). Hence, the principle of bone functional adaptation implies that bone shape, structure, and material composition are each influenced by external dynamic stresses, and thus, these features should provide a functional signal of how the bone is loaded. In other words, the principle of bone functional adaptation predicts that given that a bone is directly affected by external stimuli (e.g., lower limb bones during locomotion), the general orientation and intensity of loading could be inferred (Figure 1). This premise, if supported by experimental evidence, could be extremely beneficial in multiple fields, such as biomedical engineering, orthopedics, and the use of prostheses (e.g., reducing the negative effect of stress shielding on trabecular bone in surgical plates), space medicine (e.g., decreasing bone loss during prolonged space flight), preventative medicine (e.g., the use of in vivo HR-pQCT to predict osteopenia and osteoporosis in middle age adults), and physical anthropology (e.g., deciphering the locomotion behavior of extinct species and when our ancestors became bipedal, where all we can rely on is the incomplete fossilized remnants of the bony skeleton).

A number of previous reviews have tackled bone modeling, bone remodeling, and bone functional adaptation [4,5,6,7,8,9,10,11,12,13,14,15,16]. Yet, there has not been a literature review that specifically delves into both cortical and trabecular bone modeling and their implications for bone functional adaptation. Several reviews focused on the mechanotransduction aspect of bone modeling, namely the cellular mechanism wherein physical forces are converted into biochemical and biological responses [6,9,12,13]. Other reviews concentrated on specific aspects of bone modeling, such as influencing factors on bone modeling [11], the outcome of unloading [4], stress fractures [7], and the effect of age [13,14,15]. A handful of reviews focused on bone functional adaptation with little to no discussion of bone modeling [8,16]. Furthermore, these reviews were very restrictive in scope, addressing bone functional adaptation in specific relation to the fossil record [8] or load predictability and safety factors [16]. Finally, several reviews focused solely on bone remodeling [5,10].

While these reviews serve their intended purposes, they lack a true discussion of bone modeling and its implications for bone functional adaptation. Moreover, all of them are at least 5–10 years old, meaning they fail to encompass the latest literature on the subject. Another constraint of these reviews is their predominant focus on cortical bone modeling, with very few reviews addressing trabecular bone modeling. To date, no review has explored the similarities and differences in bone modeling between cortical and trabecular bone. Yet another drawback is that all these reviews discuss various bones, making it challenging to compare modeling responses across different studies. Finally, most reviews focus on human studies with additional information collected from murine studies, while very little information is collected on large animals.

The purpose of the current review is to remedy the abovementioned shortcomings, therefore bridging a gap in the existing literature and enhancing our comprehension of cortical and trabecular bone modeling and their implications for bone functional adaptation. It will cover recent literature and will focus solely on the mammalian tibia. Furthermore, the literature review will incorporate studies on large animals, thus giving us a more comprehensive view of the bone modeling phenomenon.

The present review will examine and discuss experimental work that shows evidence of bone modeling and supports the principle of bone functional adaptation in the mammalian tibia. The review will focus on the leg, the part between the knee and the foot, also known in humans as the crus, which consists of two bones: the tibia and fibula. However, since the fibula is non-weight bearing in humans and practically in all quadrupedal mammals (except when it is fused to the tibia), the literature reviewed, and hence the discussion presented, focuses on the tibia. In addition, to concentrate on work closely related to humans, this review will present only studies done on animals from the class Mammalia (i.e., excluding studies on birds and reptiles).

Since mice and rats are the most common animal models for skeletal human studies, most of the data in the literature are derived from murine studies. As mice and humans share highly conserved genes and molecular pathways affecting skeleton development and physiology [17], mice are considered the preferred animal model for bone modeling and bone functional adaptation studies. Nevertheless, whenever possible, studies using large animals and even humans are reviewed to demonstrate that bone adaptation to loading is a ubiquitous phenomenon in mammals. It is pointed out that studies that used human subjects are reviewed separately from those in which other animals were used due to noteworthy differences in the way these studies were designed. Specifically, human studies are less invasive, have more heterogeneous metadata (i.e., some personal information is not measured directly but is rather gathered from the test subjects via questioners), and, thus, have many more uncontrolled variables (e.g., food intake, level of physical activity etc.). In addition, this review will examine bone modeling and functional adaptation of tibial cortical and trabecular bone in distinct sections. A separate section is dedicated to discussing cortical and trabecular bone collectively. This distinction is important because previous studies have demonstrated that due to the larger surface-to-volume ratio of trabecular bone tissue, bone modeling and bone remodeling act differently in trabecular bone compared to cortical bone (except for, possibly, endocortical surfaces) [18,19,20,21,22]. Furthermore, recent studies have revealed differential gene expression and unique transcriptional profiles in cortical and trabecular bone tissues from mechanically loaded murine tibiae models [23,24], indicating that bone modeling and, thus, bone functional adaptation may be mediated differently in these two bone tissues. Similarly, it is important to mention that many of the experiments were performed using young, still-growing animals and that some studies have shown that adult or old individuals demonstrated reduced bone modeling in response to loading [25,26,27,28,29,30]. Finally, it is acknowledged that some aspects of bone modeling are still not fully understood, which underscores the complexity of the relationship between form and function. One of these open questions is related to which property or properties of mechanical stimuli affect bone adaptation to load (Appendix A, see online Appendix A). Possible properties include mechanical stimuli magnitude (peak stresses and strains) [20,28,31,32,33,34,35,36,37], sign (compression, tension, or shear) [38,39], duration (number and length of loading cycles) [31,39,40,41], circadian rhythm (i.e., time of the day) [42], strain rate (change in strain with respect to time, e.g., running vs. walking) [43], frequency (number of loading cycles per second) [32,44,45], and length of rest in between stimuli [41,46,47,48,49].

## 2. Influencing Factors in Bone Modeling in Response to Mechanical Loading

The first two aspects that need to be addressed when discussing bone modeling are that bone responds to dynamic but not static loading and that bone response to load is local, affecting only the directly mechanically stimulated bone (e.g., tibia) [45,50]. Gross et al. [51] have clearly demonstrated that the tibiae of C57BL/6J (B6) mice respond to dynamic, but not static, in vivo loads by bone deposition that yields an increase in cortical bone area. In addition, numerous studies in rats and mice demonstrate that tibial loading either by exercise (e.g., treadmill running) or in vivo loading (axial compression or four-point bending) induced bone formation only in the trabecular and cortical bone tissues of the loaded tibia but not in the contralateral non-loaded tibia [37,39,52] or in other distant non-loaded bones, such as lumbar vertebrae [53]. These findings, when considered together, suggest that mechanical stimuli signals inferred from the tibial structure are directly related to the dynamic loading of the tibia (and not another bone) during locomotion (and not during static loading such as stance).

A third key aspect of bone modeling is the effect of age, as interstitial growth via the growth plate is only active before adulthood. While cortical and trabecular bone modeling is supported by results from many experiments, it is important to point out that most of these studies were performed on young animals [37,54] or children and adolescent human subjects [55]. This information is potentially important because, in all these cases, interstitial growth via the growth plate is still active, as opposed to appositional growth (increasing the width of a bone) that continues throughout life [1]. The responsiveness of bone tissue to mechanical stimuli was shown to be augmented in actively growing animals compared to fully grown animals and adult individuals [14,25,28,29,56]. Results from these studies demonstrated that the increase of mechanical stimuli (e.g., physical activity) in young, growing animals induces bone deposition and the gaining of bone mass. Fewer studies, however, investigated bone modeling in fully grown animals and human adults. The overall findings of these studies show that bone modeling in adults and mature animals still takes place, albite at a reduced rate [2,25,26,27,28,29,56,57,58]. Nevertheless, while this decrease in responsiveness to mechanical stimuli with age has direct implications for the ability to infer changes in loading magnitude and direction of adults and mature animals, it is relevant when studying the locomotion behavior of an extinct species or in the deduction of physical activity differences between two distinct human populations, as the onset of these loads start at a young age and the loading signal was shown to be retained in adults [22].

A fourth aspect of bone modeling is the a priori impact of the underlining genetic “blueprint” on bone structure and responsiveness to external mechanical stimuli. For example, several studies investigated the response of tibial cortical and trabecular bone tissues to an increase in mechanical loading (in vivo four-point bending, jumping exercise, or low-level mechanical vibration) or unloading (sciatic neurectomy) in C3H/HeJ (C3H) and C57BL/6J (B6) mice [59,60,61,62]. While both inbred strains have similar adult body size, weight, and bone size, C3H mice have greater peak bone density compared to B6 mice. These studies demonstrated that even though both mice strains were subjected to the same loading/unloading treatment, C3H mice tibiae were less sensitive to mechanical loading or unloading compared to B6 mice, and, thus, their tibiae gained or lost less bone tissue, respectively. However, as Wallace et al. [63] pointed out, the difference in response to mechanical stimuli between two genetically identical inbred mice strains is equivalent to studying the differences in response to mechanical stimuli between two individual humans. Since paleontology studies are on entire species and paleoanthropology studies are on entire human populations, the intra-population variance in tibial response to mechanical stimuli is much smaller compared to the inter-population variance [64]. Furthermore, Holguin et al. [65] showed that variances in the duration and frequency of mechanical stimuli, as well as the length of rest in between stimuli, have far more influence on tibial bone modeling than differences between mice strains.

A further property that is related to the genetic “blueprint” of bone shape and the responsiveness to mechanical stimuli is the distinction between cortical bone and trabecular bone. Recent studies have revealed differential gene expression and unique transcriptional profiles in cortical and trabecular bone tissues from mechanically loaded murine tibiae models [20,23,24], indicating that bone modeling may be mediated differently in these two bone tissues. Additional studies have demonstrated that bone modeling is usually more extensive and happens faster in trabecular bone compared to cortical bone (except for, possibly, endocortical surfaces), possibly due to the larger surface-to-volume ratio of trabecular bone tissue [18,19,21,22]. Together, these data suggest that trabecular bone in the extremities of long bones may hold a more discerning functional signal compared to cortical bone.

## 3. Cortical Bone Modeling

The long bones of mammals are macroscopically composed of an outer dense cortical bone shell (called the cortex) and an inner porous architecture (called the trabecular bone or cancellous bone). While previous studies have established that trabecular bone is an important contributor to the mechanical behavior of whole bones [66,67,68], cortical bone is the main load-bearing component [69]. Therefore, most studies on bone modeling of the tibia focused on cortical bone modeling in response to load. Another reason for the preference of the cortical component over the cancellous component is that it is much easier and cheaper to X-ray the tibia or use peripheral quantitative computed tomography (pQCT) to capture cortical 2D cross-sectional area and cortical bone volume than to perform micro-CT scan of the tibia and analyze its 3D trabecular structure.

### 3.1. Murine Studies

The bulk of studies that investigated tibial cortical modeling used mice and rats as the animal model (Appendix A). Several of these studies loaded the whole tibia, in vivo, along its axial direction, between the knee and the ankle (Figure 2), and found that the response of the cortical component (changes in bone formation rate, cortical area, and cortical thickness) was dependent on the magnitude of load [20,33,35,37,42]. Several other studies applied four-point and cantilever bending to the in vivo tibia, which placed the medial surface of the cortex in compression and the lateral surface of the cortex in tension (Figure 2) [34,39,51]. These studies found a similar increase in cortical area (i.e., cortical modeling) in response to load. Yet another group of related studies investigated tibial cortical bone modeling in response to in vivo axial loads in mice of various ages (cyclic loading of peak force ranging between 7 and 14 N) [19,23,26,27,28,29,41,58,65]. These studies found a cortical bone modeling response (i.e., increase in cortical volume, cross-sectional area, and thickness) in the tibia of mice from all age groups. An interesting and important finding was that while cortical bone formation was greatest in young growing mice, it was still noticeable in adult mice (12 months of age) and even in aged mice (22 months of age). Chen et al. [25] applied treadmill exercise to 14-month-old female Sprague–Dawley rats. They found an increase in tibial diaphyseal mineral density, mineral apposition rate, and cortical bone formation rate in the exercised rats compared to the sedentary group.

### 3.2. Large Mammals’ Studies

Only a few studies have investigated tibial cortical bone modeling in large mammals (Appendix A). In a series of related studies, Liberman et al. measured cortical bone modeling in several hindlimb bones, including the tibia, of male Dorset sheep that were exercised on a treadmill and compared it to that in a sedentary group (control) [56,57]. In their study, the sheep were divided into 3 groups by age—juvenile, subadults, and adults. Their results demonstrated a significant difference in tibial diaphyseal cross-section area, geometry, periosteal area formation, and periosteal modeling rate between the exercised and sedentary sheep from the juvenile and subadult groups. Similar differences also existed between the exercised and sedentary sheep from the adult group, but these differences were not statistically significant. The authors stated that as an animal ages, moderate levels of exercise (loading) did not stimulate cortical modeling rates significantly. In an additional study from this group, Wallace et al. [70] used calcein-injected Dorset sheep to compare tibial periosteal bone formation between exercised (treadmill) and sedentary groups of juvenile sheep. This study also demonstrated that exercise led to a significant increase in periosteal bone formation in response to loading. However, the areas that experienced the highest compressive and tensile strains during peak loading did not correspond to the areas that demonstrated the largest periosteal bone formation. These findings confirm the presence of bone modeling in response to load but are not consistent with the expected outcomes of bone functional adaptations.

A different approach was adopted by Niinimäki et al. [71], who conducted a study of 3 groups of modern reindeer—free-ranging/wild reindeer (served as reference) racing and draft (working, increased exercise) and zoo dwellers (sedentary). They tested the hypothesis that habitual loading patterns and, thus, cortical modeling may differ between domesticated and free-ranging/wild reindeer due to human influence. This study found differences in activity-modified bone cross-sectional properties and external dimensions of long bones, and, specifically, the tibia, which indicated an improved long bone robusticity in response to increased loads in working reindeer.

The main finding of the abovementioned studies [56,57,70,71] was that increased loading of the tibia generated a cortical bone modeling response that increased cortical bone thickness, area, and mass. Consequently, animals that experienced higher levels of activity tend to demonstrate adapted tibial morphology that is better in resisting bending (predicted from the ratio between the maximum and minimum second moments of area (I_max_/I_min_)) and torsion (predicted from the polar moment of area (J)).

### 3.3. Human Studies

Studies that explore cortical bone modeling in humans are very important, but they are limited in scope due to ethical issues and safety concerns (e.g., exposure to radiation during bone imaging). In a study of about 200 school children from the British Columbia School District (4th and 5th grade, average age 10.2 years), Macdonald et al. [55] tested the effect of exercise (jumping, skipping, dancing, and playground circuits) on the tibial mid-diaphysis cortical area, thickness, and second moment of area of the children. Using pQCT on day zero (baseline) and after 16 months, these researchers reported a significant increase in the second moments of the area of the tibia shaft (a proxy for diaphyseal bending strength) in the exercised group. Cortical area and thickness were also greater in the exercised group, but the difference was not significant. Murray and Erlandson [22] also studied the effect of physical activity on cortical bone modeling in school children. They examined tibial mid-diaphysis cortical parameters in young adult females (aged 19–33 years) who were either sedentary or active (gymnasts or runners) pre-puberty and were now, post-puberty, either sedentary or active. Using pQCT at the proximal tibial diaphysis, these researchers found significant increases in cortical cross-sectional area (i.e., bone modeling) and strength-strain index (a measure of bending and torsional strength derived from section modulus and cortical bone density) in all the adults who were active pre-puberty compared to sedentary. The increase in these cortical parameters was regardless of the post-puberty level of activity of either group.

Two recent studies investigated the effect of physical activity in adults on tibial cortical bone modeling [21,72]. Saers et al. [21] measured tibial distal diaphysis cross-sectional area and resistance to bending (I_max_/I_min_) and to torsion (J) in 88 adult males (age range of 19–30 years) who were sedentary, low-impact loading athletes (swimming) or high-impact loading athletes (running, hockey and cricket). They found an increase in cortical bone parameters in high-impact-loading athletes compared to sedentary adults and low-impact-loading athletes. Similarly, Hughes et al. [72] found a significant increase in cortical thickness of the distal tibia (pQCT) after 8 weeks of basic Army combat training in both sexes and various racial groups (n = 1605, age range of 17–42 years). A different aspect of tibial cortical bone modeling, namely bone resorption due to disuse, was studied by Gabel et al. [3]. The distal tibiae of 17 astronauts (average age of 47 years) were scanned using high-resolution peripheral computed tomography (HR-pQCT) before and 12 months after their space mission. The results revealed that the median estimated tibial strength (failure load) and cortical bone mineral density did not recover fully and stayed low even 12 months post-space flight.

### 3.4. Summary

Although the combined results of these studies clearly illustrate a cortical bone modeling response to loading, only a few studies have reported on the ramifications of cortical bone modeling for the principle of bone functional adaptation. These studies have shown that tibial cortical bone functional adaptation does not always correspond to the expected bone surfaces with the highest stresses and strains [39,70,73,74,75]. Lieberman et al. [74] found that the I_max_ in sheep tibiae that were exercised on a treadmill did not agree with the plane in which the bone was bending. Wallace et al. [70] found that while treadmill running produced peak strains in the anterior and posterior cortices of sheep tibia (tensile and compressive, respectively), cortical bone formation (i.e., modeling) occurred mostly at the medial cortex. Javaheri et al. [75] also noticed a cortical bone modeling response, which was independent of local strains, in the cortex of tibiae from C57BL/J6 female mice subjected to dynamic load for 2 weeks. Thus, these results demonstrate that while tibial cortical modeling occurs in response to increased loading, this does not always correlate with the actual bone surfaces that endure the highest stresses and strains (as predicted from the concept of bone functional adaptation). A possible explanation for this unexpected outcome was suggested by van Tol et al. [76], who mapped the architecture of the lacunocanalicular network in mice tibiae after 2 weeks of mechanical loading. They found that contrary to local strain, calculated fluid flow through the lacunocanalicular network is an excellent predictor of bone formation on the tibia periosteal surface. As the flow of lacunocanalicular fluid is the primary source of shear stresses on the osteocyte cell processes within the canaliculi, the strains measured locally on the cortical bone surface may be unreliable, leading to a misidentification of the areas with the highest stresses and strains. A different explanation was given by Miller et al. [77]. They also noticed that tibial cortical bone formation in response to loads did not necessarily occur in regions of the highest compressive and tensile strains. Thus, they hypothesized that bone formation in response to load depends not only on load magnitude but also on the specific bone tissue position along the periosteum surface. Consequently, studying tibial cortical bone adaptation and tibial resistance to bending (I_max_/I_min_) and torsion (J), based solely on local strains and stresses, may not be helpful in identifying areas that underwent the most modeling and the magnitude and type of mechanical stimuli. This prospect is supported by Shaw and Stock [78], who showed that human tibial rigidity measurements do not differentiate between adult long-distance runners, who started practicing during adolescence and have been competing for about 10 years, and adult non-runners.

## 4. Trabecular Bone Modeling

Similar to tibial cortical bone modeling in response to load, tibial trabecular bone also adjusts and models in response to loads by gain in bone mass (i.e., increase in trabecular bone volume fraction (BV/TV)) and changes in trabecular architecture (e.g., changes in trabecular thickness (Tb.Th), number (Tb.N), and degree of anisotropy (DA)).

### 4.1. Murine Studies

A large body of literature described an experimental approach that involved the application of noninvasive in vivo axial loads to mice tibiae (Appendix A). While the studies varied in terms of the parameters used, such as the magnitude of load (3.5–13 N), length of experiment (1–6 weeks), sex (male/female), age (8 to 26 weeks), and the region of the tibia that was analyzed (proximal/distal), the vast majority of them reported a similar trend in the results [20,26,28,29,33,35,37,41,42,52,53,58,65,79,80]. This was that the trabecular modeling response of increased BV/TV happened through the increase of Tb.Th (but generally not Tb.N), in response to loading. However, results reported by Grimston et al. [23] and some results in De Souza et al. [33] and Brodt and Silva [58] did not show the same finding.

Berman et al. [37] applied 9 cyclic, noninvasive, in vivo, axial compressive loads to the right tibiae of 30 8-week-old C57BL/6 mice over 2 weeks. The mice were divided into 3 groups (10 mice per group), and each group was subjected to a different maximum compressive force of 8.8 N, 10.6 N, or 12.4 N. Micro-CT scans of the right and left proximal tibiae revealed a dose-dependent increase in trabecular BV/TV and Tb.Th, as well as in trabecular tissue mineral density (TMD), in the loaded right tibiae in comparison to the non-loaded, control, left tibiae. This and similar studies demonstrated that lower magnitude of loads, such as 3.5 N [20,35], and shorter trial periods, such as 1 week [58], tended to induce weaker trabecular modeling responses. Furthermore, older mice usually demonstrated reduced trabecular bone modeling in response to load [28]. An interesting detail that emerges from these studies is that the increase in trabecular BV/TV in response to mechanical stimuli is primarily due to an increase in Tb.Th. The fact that trabecular BV/TV increased due to an increase in Tb.Th and not Tb.N (even in young, growing mice with active growth plates) provides support for the postulate that trabecular bone responds to an increase in loading mostly by new bone deposition (i.e., bone modeling) and less by the formation of new trabeculae. This is an interesting outcome as it appears to be inconsistent with a previous finding that rodents and humans reveal two distinct mechanisms to achieve variations in trabecular BV/TV [81]. Barak et al. [81] noted that while variation in Tb.Th is the main contributing factor for differences in trabecular BV/TV in humans, and variation in Tb.N is the main contributing factor for differences in trabecular BV/TV in rodents (mice and rats). However, this finding refers to baseline differences in how genes control trabecular BV/TV between species and not due to a modeling outcome in response to mechanical stimuli within a species.

### 4.2. Large Mammals’ Studies

As is the case with cortical bone, there are also only a few studies that involved the use of large mammals to investigate trabecular bone modeling (Appendix A). This is possibly due to the difficulty of exercising a large enough number of large mammals for a sufficient duration to elicit a substantial and detectable trabecular modeling response. In addition, compared to mice and rats, large mammals require extensive housing, care, and feeding, which are usually expensive. One way to overcome this problem is to study the skeleton of genetically related species that differ in their amount of habitual daily activity (i.e., locomotion) and, thus, differ in the mechanical stimuli their bones experience. This approach was adopted by Assif and Chirchir [82] in a study on 4 different species from the felid family, each with a different daily travel distance and home range size. They have used pQCT to scan a total of 50 distal tibiae, as well as a similar number of femoral and humeral heads, from 4 large mammals, namely leopards, mountain lions, jaguars, and cheetahs. The results showed that even though jaguars travel the longest distance daily, they had the lowest trabecular BV/TV in the distal tibia (no significant difference was found in the femoral and humeral heads). While these results appear to contradict the principle of trabecular bone functional adaptation, it should be noted that this study has 2 limitations in relation to bone adaptation. First, as different mammals were compared, the contribution of genetic differences to trabecular bone architecture must be considered; however, this was not done in the study. Second, in the study, the activity of living animals was not regulated; rather, bones from museum collections were used, many of them with unknown origin, and the daily travel distance values reported were means taken from the literature and not the actual recorded activity of the study animals. Thus, the approach in the study by Assif and Chirchir [82] should be used carefully to assess bone functional adaptation.

Another way to overcome the difficulties mentioned above in using large mammals is to reanalyze older data from previous studies with a new and improved tool or technique. Cazenave et al. [83] reanalyzed data from both the left and right proximal femur and proximal tibia of a male Japanese macaque that underwent bipedal locomotion exercises 30–60 min per day for 8 years [83,84]. While the new analysis did not find differences in global trabecular parameters (e.g., trabecular BV/TV, Tb.Th, and Tb.N) between the bipedal Japanese macaque and the 5 other wild animals, there was a difference in the trabecular architecture of both the proximal femur and tibia. Compared to the wild animals, the proximal tibia of the bipedal macaque demonstrated a slightly more medially placed bone reinforcement in the lateral condyle and a slightly higher trabecular DA in the medial compared to the lateral condyle. DA estimates how well trabeculae align along a preferred axis. It is measured using the mean intercept length (MIL) technique. A linear grid is superimposed on a selected area (2D) or volume (3D), and then we count how many intersections happen between the grid and the bone/non-bone interface. MIL is defined as the total length of the line divided by the number of intersections. Next, the grid is rotated at a constant predetermined angle, and the counting takes place again. By repeating this measurement numerous times, we can determine at what orientation the MIL is the smallest or largest (it has the greatest or fewest number of intersections between the grid and the bone/non-bone interface, respectively). DA equals 1 minus the sum of the minimum value divided by the maximum value. Thus, DA will range between zero and one, where 0 represents total isotropy, and 1 represents total anisotropy. This switch implies that the trabecular architecture refines itself into a more mechanically efficient, anisotropic structure. Interestingly, the differences reported in this study [84] agreed with similar findings, which were noted previously in the corresponding distal femur of the bipedal macaque. The finding of this study suggests that global trabecular parameters, such as BV/TV, Tb.Th, and Tb.N may be adequate to detect a general bone modeling response to increased mechanical stimuli, but they may not be discerning enough to detect a functional adaptation response to changes in locomotion behavior. During habitual locomotion behavior, the bone is loaded, on average, in a predicted way. Under these loads, trabecular bone surfaces that tend to experience low stresses and strains will be resorbed, and surfaces that tend to experience high stresses and strains will experience bone deposition. The long-term sum effect of these two independent events, occurring on different trabecular surfaces, is the change in trabecular architecture and principal trabecular orientation to better accommodate this loading direction.

In the case of habitual locomotion, tibial trabecular functional adaptation starts as soon as the young animal begins locomoting [85]. Tanck et al. [85] studied the proximal tibia of pigs from 5 different age groups (6, 23, 54, 104, and 230 weeks of age). They reported 2 distinct phases of trabecular bone adaptation to habitual locomotion. At a young age (6 weeks), pigs respond to physiological loading by an increase in trabecular BV/TV. Then, at 23 weeks, the trabecular adaptation response switches from an increase in trabecular mass to an adjustment in trabecular orientation (i.e., DA). Tanck et al. [85] found that as the pigs matured, the angle between the principal loading direction during locomotion and the calculated stiffest direction of the trabecular tissue decreased, suggesting that the trabecular structure aligned itself in response to habitual locomotion. In further support of these findings, Barak et al. [54] demonstrated that the trabecular architecture in the distal tibia of young sheep, which were exercised on an inclined treadmill, adjusted its orientation in response to a continuous change in their hock joint loading orientation. These findings demonstrated that trabecular bone dynamically adjusts and realigns itself in relation to changes in peak loading direction during locomotion, and thus, they support the principle of bone functional adaptation.

### 4.3. Human Studies

Two recent studies investigated the effect of physical activity on trabecular bone architecture in the human distal tibia. Saers et al. [21] reported that compared to a sedentary control group, active athletes had significantly higher trabecular bone mineral density (BMD), irrespective of the sport, which is involved in impact or nonimpact loading (i.e., swimming). These results were different from what they found for the tibial cortical bone. Similarly, Hughes et al. [72] found an increase in trabecular BMD, trabecular BV/TV, and Tb.Th in 1605 trainees, after 8 weeks of basic Army combat training, in both sexes and different racial groups. At the other end of the trabecular modeling spectrum, Gabel et al. [3] reported a significant decrease in trabecular BMD, trabecular BV/TV, and Tb.Th, in 17 astronauts, 12 months after return from space missions. The researchers estimated that this outcome is comparable to a decade or more of age-related bone loss. These studies demonstrate that trabecular bone modeling in the human tibia takes place by new bone deposition and old bone resorption, depending on the magnitude of the mechanical stimuli that the bone experiences.

Two additional human studies investigated trabecular bone modeling during childhood. Murray and Erlandson [22] found that trabecular BMD was higher in the distal tibia of 44 female athletes compared to 37 female sedentary controls (total n = 81, age range 19–33 years). These outcomes were independent of the amount of activity carried out by the athletes during childhood (8 of the sedentary adult females were athletes pre-puberty) and different from what was found for cortical bone. This was that increases in the cortical cross-sectional area were related to pre-puberty activity level, regardless of the post-puberty level of activity. This indicates that, in contrast to cortical bone, trabecular bone is highly responsive to changes in mechanical stimuli (increase or decrease), even in adult individuals. In a non-related study, Raichlen et al. [86] correlated structural changes of trabecular bone in the distal tibiae of young children (1–8 years old) to locomotor changes in the ankle joint during ontogeny. Younger children who had just started to walk exhibited kinematic instability and variation in the loading of the ankle joint. Correspondingly, these children demonstrated lower trabecular DA and a large variation in DA between individuals. In contrast, older children who had more stable locomotion demonstrated higher DA and decreased variance in DA between individuals.

Building on these and other studies, Barak et al. [87] investigated the relationship between principal trabecular orientation and ankle joint angle at the point of peak ground reaction force in humans and chimpanzees. As chimpanzees locomote quadrupedally and humans walk bipedally, chimpanzees have significantly more flexed ankle joint angles at the point of peak ground reaction force during the gait cycle. Corresponding to the difference in ankle joint angles, Barak et al. found that the principal trabecular orientation at the distal tibia of chimpanzees was significantly more oblique in relation to a perpendicular plane to the long axis of the bone in the sagittal plane (Figure 3). These researchers then used these results to infer from the calculated principal trabecular orientation of fossilized distal tibiae assigned to Australopithecus africanus that this species is locomoted like modern humans with a relatively extended posture.

### 4.4. Summary

Various studies have demonstrated active trabecular bone modeling in response to physiological dynamic loading, especially as indicated by changes in trabecular bone architecture. These changes were shown to take place primarily by increasing Tb.Th (i.e., bone deposition, or in other words, bone modeling) rather than by the creation of new trabeculae (i.e., increase in Tb.N). However, as pointed out in this present review, there are several limitations associated with relying solely on the trabecular bone component of whole bones to study bone modeling in the context of bone functional adaptation. Mainly, the complex 3D structure of trabecular bone is interconnected and integrated with the cortical bone around it. Consequently, exclusive interpretation of trabecular bone modeling is limited at best. While calculating and analyzing the principal orientation of trabecular bone has many advantages [54,87,88], the analysis of bone functional adaptation should adopt a combined approach of investigating both cortical and trabecular bone tissues, as each of them separately and together can enhance our understanding of this important phenomenon.

## 5. Integrated Cortical and Trabecular Bone Modeling Analysis

Appendix A gives an overview of studies that concurrently investigated trabecular and cortical bone modeling in the tibia. While some studies found similar modeling responses between trabecular and cortical bones [26,29,34,41,42,65], other studies found that cortical bone modeling response to loading was more significant [20,23,28,29,33,35,37,89]. Only one study, by Yang et al. [19], reported an opposite effect where trabecular bone modeling response to loading was more significant. Yet, it is important to acknowledge that bone modeling studies, whether on cortical bone or trabecular bone, primarily concentrate on murine animal models, and the relatively few large animal and human bone modeling studies are mostly retrospective, a product of data reassessment or ex post facto analysis of bones [71,82,84,85]. Only Lieberman et al. [56,57], Barak et al. [54], and Wallace et al. [70] have performed controlled experiments to test the concept of bone modeling in cortical and trabecular bone tissues in the tibia of large animals.

Discrepancies in the modeling response to loading were noted between cortical and trabecular bones concerning both the intensity and duration of loading [15,20,21], as well as the duration of retention of the modeling response. Saers et al. [21] noted that cortical bone modeling is significantly higher in sports involving impact loading (e.g., running) compared to nonimpact loading (e.g., swimming, controls), Yet trabecular BMD (a proxy for BV/TV) was higher (but not significantly different) in all athletes, regardless of their type of sport, compared to controls. These results suggest that trabecular and cortical bones are affected differently by loading magnitude. Similarly, Yang et al. [20] reported that cortical bone in the tibia midshaft of mice demonstrated a significant modeling response in all 3 loading experiments (3.5, 5.2, and 7 N), even if the response was stronger with higher applied loads (5.2, and 7 N). Yet, only at the highest loading magnitude (7 N) was a significant modeling response produced in the proximal metaphyseal trabecular bone. In another study by the same group, Yang et al. [41] found that the modeling response of trabecular bone to loading reaches saturation faster than cortical bone. The modeling response of cortical bone continues to increase from 36 to 216 cycles of loading and even up to 1200 cycles, whereas the modeling response of trabecular bone plateaus after 216 cycles of loading. The authors postulated that this difference may arise from differences in the cortical and trabecular bone biological microenvironment or that extended loading durations induce damage to the articular cartilage and subchondral bone, which could then negatively influence the underlying metaphyseal trabecular bone. Murray and Erlandson [22] identified a difference in the duration of retention of the modeling response between trabecular and cortical bones in premenopausal adult females. While tibial cortical bone modeling during childhood/adolescence was still evident in adults, whether or not they were still active athletes, trabecular bone BMD in the distal tibia (a proxy for BV/TV), was only higher in active adult female athletes regardless of their level of activity during childhood/adolescence. Together, these findings suggest that a comprehensive understanding of lifelong loading histories from bones (i.e., modeling response to loading) can be achieved more accurately by adopting an integrated approach, namely one that incorporates analyses of both cortical bone and trabecular bones.

Although the collective findings in the reviewed articles indicate a definite bone modeling reaction to loading in both cortical and trabecular bones, these findings also underscore the limitations of the isolated approach when these findings are used to explore the principle of bone functional adaptation. While cortical bone modeling may not consistently align with anticipated high-stress and strain bone surfaces [70], trabecular bone modeling is harder to interpret, and it appears to necessitate greater loads and likely diminishes more rapidly with age [20,33,35]. Various factors may account for the differences observed in the response of cortical and trabecular bones to loading. For both trabecular and cortical bones, modeling is accomplished via the action of osteoblasts (bone deposition) and osteoclasts (bone resorption). These cells, in turn, are regulated by gene activity. Both Kelly et al. [24] and Yang et al. [19] found that different genes are activated in response to loading in trabecular and cortical bones. Hence, it is likely that each of these bone tissues would exhibit distinct responses to loading. For example, this difference in gene activation may indicate that trabecular and cortical bones have different thresholds to strains, specifically, a higher threshold for loading and a lower threshold for unloading in trabecular bone [19]. Alternatively, it is conceivable that trabecular bone experiences lower strains during loading because cortical bone primarily bears the load, while trabecular bone plays a larger role in distributing and dispersing loads away from the joint surface [41,67,68]. Furthermore, it is important to acknowledge the uncertainty surrounding the loading conditions of trabecular bone. While peak strains at the cortical mid-diaphysis can be assessed using strain gauges, this method cannot be used on trabecular bone. Even if strain gauges were placed on the metaphyseal surface, they would only measure the local strains in the cortical component, which represent a combination of contributions from both cortical and trabecular bone tissues. Without directly determining trabecular bone strains, it is impossible to distinguish whether the decreased response of trabecular bone to loading in older age and under lower loads is due to differences in localized peak strains between cortical and trabecular bones, different thresholds to strains for these two bone tissues, or some other underlying cause. It is also important to recognize that compared to cortical bone, the complex 3D architecture of trabecular bone increases the difficulty of assessing its response to load. Several studies demonstrated an unpredicted decrease in BV/TV in response to loading due to a decrease in Tb.Th and/or Tb.N [23,33,58]. These results may represent the optimization of trabecular architecture (trabecular aligning along the loading direction), which then requires less bone material to withstand the same loads (stresses). Tanck et al. [85] reported this phenomenon, noting an initial increase in trabecular BV/TV in the fifth lumbar vertebra and the tibia epiphysis and metaphysis in young pigs up to around 50 weeks of age, but then, a subsequent decline commenced. Concurrently, trabecular DA began to increase at around 50 weeks, continuing until about 100 weeks of age. It is suggested here that the transition from adjusting bone quantity (the volume of trabecular bone indicated by BV/TV, Tb.Th, and Tb.N) to adjusting bone quality (the reorganization of trabeculae indicated by DA and trabecular connectivity) is similar to the synaptic pruning process observed in the brain between early childhood and adulthood. During synaptic pruning, the brain eliminates unnecessary neurons and synapses, which helps the brain to adapt and enhance its efficiency of neural transmissions [90]. Similarly, it is proposed here that trabecular bone undergoes an initial pruning process in which unnecessary minimally loaded trabeculae are eliminated (resorbed, i.e., bone modeling), which helps the bone to adapt and enhance its efficiency in transmitting loads from the joint surface to other cortical bone regions. Thus, the dual examination of cortical and trabecular bones, individually and collectively, can contribute significantly to enhancing knowledge about bone modeling and improving interpretations of bone functional adaptation.

## 6. Clinical Importance

Bone modeling plays a pivotal role in maintaining skeletal health and responding to mechanical stresses. During childhood and adolescence, bones undergo significant modeling via bone deposition to accommodate changes in size, shape, and mechanical demands. Abnormal modeling can lead to skeletal deformities such as kyphosis, highlighting the importance of early detection and intervention. On the other hand, inadequate mechanical stimulation, as seen in prolonged bed rest or immobilization, can lead to bone loss and increased fracture risk, particularly in the elderly and individuals with certain medical conditions like osteoporosis. Advancements in imaging technologies, such as dual-energy X-ray absorptiometry (DXA) and high-resolution peripheral quantitative computed tomography (HR-pQCT), allow clinicians to assess bone quality and quantity, providing valuable insights into bone modeling in various clinical settings. Additionally, clinicians can employ this knowledge to devise screening procedures for examinations of musculoskeletal health status in healthy, athletic, or diseased-state populations, as well as personalized treatment approaches for patients with skeletal disorders or those undergoing orthopedic procedures. In conclusion, one of the goals of bone modeling research is to translate scientific discoveries into tangible benefits for patients. By understanding the clinical implications of bone modeling, healthcare professionals can better manage bone-related disorders, optimize therapeutic interventions, design effective exercise regimens for improving bone health and preventing osteoporosis-related fractures, and promote skeletal health across diverse patient populations.

## 7. Limitations

This review sheds light on several gaps and limitations in our present comprehension of bone modeling and functional adaptation, necessitating further research. The primary gap lies in the fact that most studies focus solely on cortical bone modeling. There are fewer examinations pertaining to trabecular bone modeling, and even fewer studies encompass both of these bone tissues. Yet, as previously explained, there is a clear indication of disparities between cortical and trabecular bone modeling responses, along with their implication for bone functional adaptation.

A second related issue is that, in contrast to our existing knowledge and assumptions, cortical bone modeling does not always correspond to the expected bone surfaces, which seem to experience the highest stresses and strains. These findings indicate that our understanding of cortical bone modeling remains incomplete, which is clear from the plethora of potential influencing factors suggested for bone modeling (Appendix A). This also highlights the fact that by using strain gauges, we only approximate the strains in specific directions and locations on the cortical bone surface. Furthermore, these measured strains represent the collective summation of strains within both the cortical bone and the underlying trabecular bone tissue. Therefore, determining accurately or directly quantifying site-specific cortical or trabecular strains and stresses remains equivocal at best.

Finally, there is presently a particular technical constraint regarding image resolution when attempting to implement our comprehension of trabecular bone modeling in clinical settings. Current in vivo image resolution, especially in humans, is limited. Second-generation HR-pQCT with a voxel size of 61 μm appears to be pushing the boundaries of capturing in vivo trabecular bone resorption and formation. A much higher image resolution will be needed to enable us to use our bone modeling and functional adaptation information to optimize personalized therapeutic interventions and treatments.

## 8. Future Studies

To further our understanding of bone modeling and its implication for bone functional adaptation and to leverage this knowledge to advance preventative and personalized medicine, more research is required in human and murine animal models, as well as in large mammals. Mechanotransduction, the process by which mechanical forces are converted into biochemical signals, lies at the heart of bone modeling. Future studies into cortical and trabecular bone modeling should prioritize the integration of multiscale data, spanning from molecular, cellular, tissue, and organismal levels. Omics technologies, such as genomics, transcriptomics, and proteomics, can shed light on the genetic and molecular factors influencing bone modeling in health and disease and elucidate the intricate signaling pathways that regulate bone modeling in response to mechanical stimuli. One promising avenue for future exploration is the study of gene and protein expression in osteocytes engaged in the mechanoregulation of bone modeling, encompassing both resorption and formation processes. This will help us pinpoint which factor or factors influence cortical and trabecular bone modeling. When combined with cutting-edge imaging techniques, such as time-lapse in vivo micro-CT/HR-pQCT, micro-FE analysis, and biomechanical tests, these data can help explain the complex regulatory mechanisms governing cortical and trabecular bone modeling and adaptation across scales. Initial steps have been made in this direction [20,23,24,50,62,89]; however, these efforts must continue and expand to generate substantial and significant data from which meaningful conclusions can be derived. Future studies should particularly focus on elucidating the similarities, disparities, and interactions between cortical and trabecular bone modeling, as the existing data suggests that understanding these aspects is paramount for clinically applying this knowledge.

Another aspect of bone modeling that should be addressed in future studies is the fluid flow through the lacunocanalicular network. Past research has shown that bone modeling may not be solely dictated by strain magnitude [39,70,73,74,75], with fluid flow through the lacunocanalicular network proposed as an additional factor influencing bone formation and resorption [76]. As the flow of lacunocanalicular fluid around the osteocyte cell processes is the chief inducer of shear stresses, it has been proposed that strains measured on local cortical bone surfaces may not accurately reflect the strains perceived by osteocytes. This discrepancy could lead to misidentification of the bone regions experiencing peak stresses and strains.

Yet another promising avenue for future exploration is the study of bone modeling in relation to ontological development. Drawing from prior findings [86], this author hypothesized that akin to synaptic pruning at a young age, trabecular bone initially undergoes a pruning phase, during which expendable minimally loaded trabeculae are eliminated. Following this initial phase, existing trabeculae will undergo bone adaptation, involving bone deposition and/or resorption, to modify the 3D trabecular architecture. To further explore this intriguing idea, future studies will need to study trabecular bone modeling in a series of ontogenic studies using time-lapse in vivo micro-CT or HR-pQCT scanning. Finding support for this hypothesis may help explain the diverse and sometimes conflicting results various studies have presented.

Finally, while this review and much of the preceding research focus on bone modeling in mammals, it is valuable to extend the exploration of bone modeling to other vertebrates. For example, while the prevailing mechanotransduction paradigm indicates osteocytes as the cells responsible for detecting local strains and initiating bone modeling, it is important to recognize that over 50% of all vertebrate species are teleost fish, which lack osteocytes (i.e., acellular bone) [91,92]. Nonetheless, they still undergo bone modeling [93]. Understanding the mechanism of bone modeling in acellular bone will undoubtedly enhance our comprehension of bone modeling in mammals.

## 9. Conclusions

This review aimed to address a gap in the existing literature regarding bone modeling and its implications for bone functional adaptation. For the first time, both cortical bone modeling and trabecular bone modeling were assessed both individually and collectively. Moreover, the entirety of the review concentrates on a particular bone, specifically the tibia, therefore enabling readers to directly compare the findings of various studies. Finally, this review addresses not only humans and murine animal models but also large animals, therefore offering a more comprehensive overview of the bone modeling phenomenon across mammalian species.

As depicted in the review, the experimental evidence supporting cortical and trabecular bone modeling in the tibia may be summarized in six key points. First, the literature shows clear support for cortical and trabecular bone modeling in general and specifically in the tibia. Second, the mechanical stimuli that trigger bone modeling must be dynamic. Third, the mechanical stimuli must be above a certain threshold (magnitude or duration) to trigger bone modeling. Fourth, the mechanical stimuli trigger a local bone modeling response (i.e., only affecting the directly loaded bone(s)). Fifth, bone modeling in the tibia and other bones is more pronounced in growing animals and children but is usually still evident in adults. Sixth, while it appears that cortical bone responds in a more general way (increase in bone mass and cross-sectional area), trabecular bone architecture and, specifically, principal trabecular bone orientation appear to capture the habitual loading direction more exactly. One possible explanation for this postulate is that cortical bone is stronger compared to trabecular bone as it is the fundamental mechanical component that endures loading. On the other hand, trabecular bone functions to dissipate and distribute the loads that are transmitted across the joint (in the case of the tibia—the knee and ankle joints) and, as such, trabecular bone is more responsive to locomotion signals during habitual loading. It is suggested here that an integrated approach to studying both external (cortical) and internal (trabecular) bone structure and architecture would illuminate the interpretation of bone modeling in the context of bone functional adaptation. Recent developments, such as the growing use of micro-CT scanning (including in vivo micro-CT scanning in humans [3,72]), additive manufacturing (3D printing) of trabecular bone samples [94,95], and advanced computational and numerical methods [20,96,97,98,99], should facilitate the use of this approach, culminating in improvements in fields that range from the care of osteopenic elderly patients to the study of locomotion behavior in extinct species.

## Figures and Tables

**Figure 1 bioengineering-11-00514-f001:**
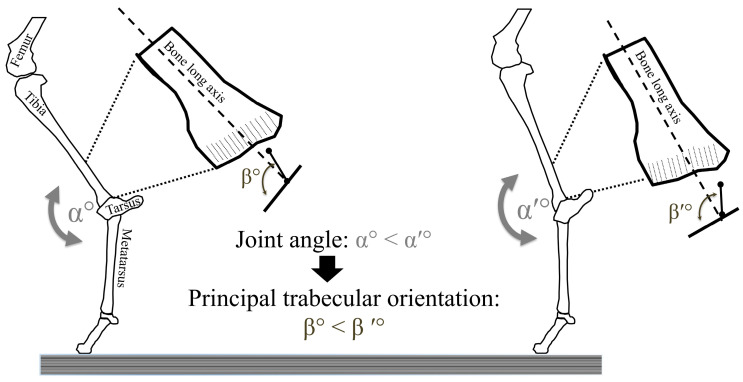
A simplified generalized schematic 2D representation to explain the principle of bone functional adaptation in response to a change in habitual loading of the ankle joint (i.e., not actual experimental results). A habitual change in ankle joint angle (from α to α′) during peak ground reaction force will, with time, induce an adjustment in the trabecular bone structure of the distal tibia. This bone functional adaptation response will change the principal orientation of trabeculae in that location (from β to β′).

**Figure 2 bioengineering-11-00514-f002:**
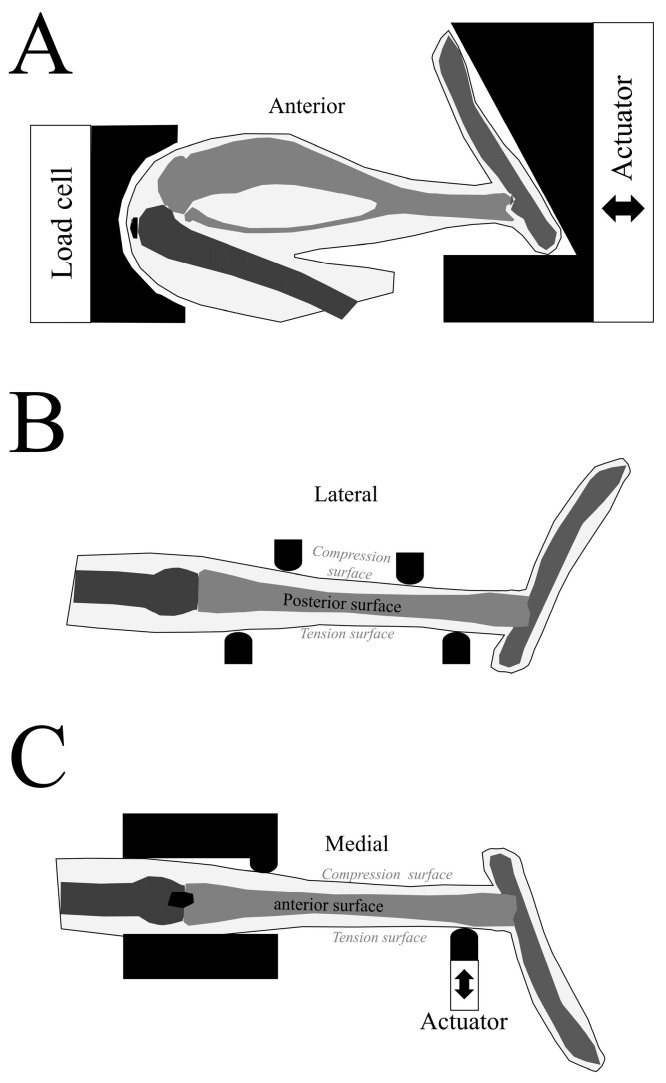
Different approaches of in vivo loading the murine tibia. (**A**) Axial compression. The leg (tibia and fibula) is constrained between a proximal padded loading cup (knee) and a distal padded loading cup (ankle). The distal loading cap is connected to an actuator that applies axial loading to the tibia (note—in some experiments, the location of the actuator is switched to the proximal loading cup). (**B**) Four-point bending. The medial aspect of the leg is placed on supports while a pair of actuators applies a load on the lateral side of the leg. This generates compressive stresses on the lateral side of the tibia and tensile stresses on the medial side of the tibia. (**C**) The distal thigh and the proximal leg (femur and tibia/fibula, respectively) are fixed, and loads are applied to the lateral aspect of the distal tibia via an actuator. This generates a tensile stress on the lateral side of the tibia and a compressive stress on the medial side of the tibia.

**Figure 3 bioengineering-11-00514-f003:**
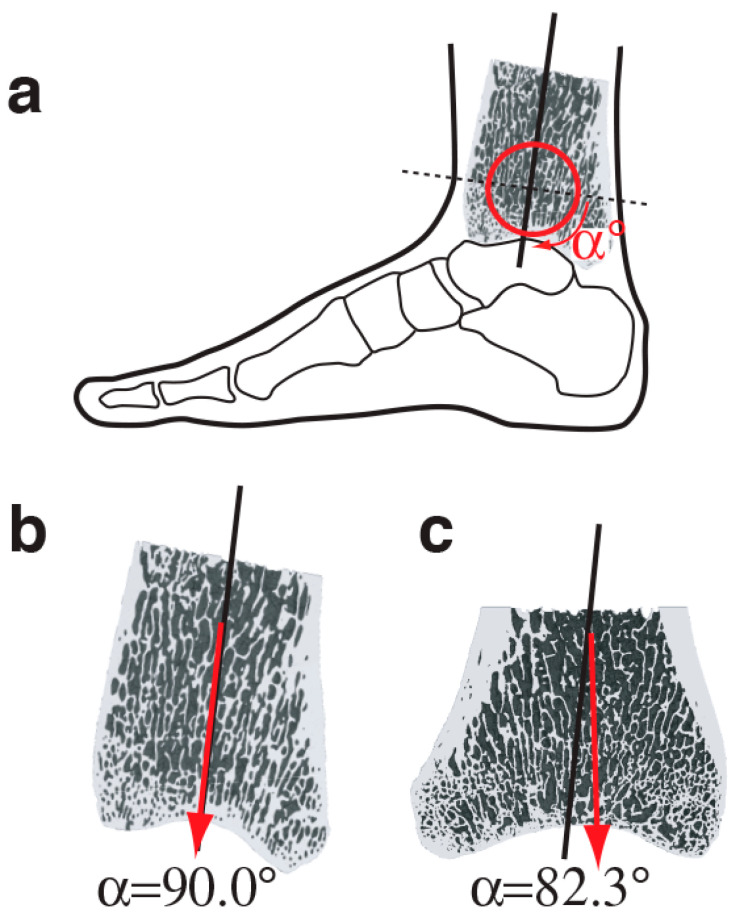
Mid-sagittal views of the distal tibia acquired from micro-CT scans. The anterior aspect corresponds to the left side of each bone. Vertical inclined black lines represent the long axis of the bone. The principal trabecular orientation for each bone is represented by a vertical inclined red arrow ((**b**) for humans and (**c**) for chimpanzees). The principal trabecular orientation is defined as the angle (α) between the principal trabecular orientation and the perpendicular plane to the long axis of the bone (represented as a horizontal dashed line in (**a**)). Adapted from Barak et al. [87] under the Creative Commons Attribution (CC BY) license.

## Data Availability

Data are contained within the article and Appendix A.

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
