# Peer review of "Cortical and Trabecular Bone Modeling and Implications for Bone Functional Adaptation in the Mammalian Tibia"

_bioengineering, 2024, doi:10.3390/bioengineering11050514_

Round 1
Reviewer 1 Report
Comments and Suggestions for Authors
The review article written by Barak et al. on the title of “A review of cortical and trabecular bone modeling and the implication to bone functional adaptation, using the mammalian tibia as a case study”.
Minor Revision Suggestions:
1. Clarify the terminology: In the introduction, provide a brief explanation or definition of terms like "cortical" and "trabecular" bone tissues for readers who may not be familiar with them.
2. Provide context for the significance: Expand on why understanding bone modeling and functional adaptation is important for orthopedic solutions and preventive medicine. How does it impact patient care or treatment strategies?
3. Include specific examples: While discussing the literature, incorporate specific studies or findings to illustrate the points being made. This could enhance the credibility and relevance of the review.
4. Address potential limitations: Acknowledge any limitations or gaps in the current understanding of bone modeling and functional adaptation. Are there areas where further research is needed?
5. Consider broader implications: Explore how the findings of this review might extend beyond mammalian biology. Are there implications for understanding bone adaptation in other vertebrates or even in engineering applications?
6. Improve flow and organization: Ensure a smooth transition between sections, from the introduction to the literature review and conclusions. Clear subheadings can help guide readers through the article's structure.
Reviewer 2 Report
Comments and Suggestions for Authors
The article “A review of cortical and trabecular bone modeling and the implication to bone functional adaptation, using the mammalian tibia as a case study” reviews articles on bone remodelling problems in the mammalian (including human) tibia. Remodelling of cortical and trabecular bone under external load was reviewed separately for different mammals such as mice, sheep, or humans from childhood to the elderly age. The bone remodelling is well known and before current investigation, but systematic investigation of articles of different authors shows that a bone remodelling requires specific threshold load conditions to occur. That is new to me and may be new to other readers, too. The only remark I have is the question about using pictures (Figure 3) of other researchers in the paper without citing it in the reference list and mentioning the appropriate license for use. Other things like article structure, section titles, or content inside them are logical and clear. Therefore, my proposal is: the article can be published if the publishing ownership rights of all figures are ok.
Reviewer 3 Report
Comments and Suggestions for Authors
Please see the uploaded ZIP FILE, which contains 2 Word documents.

Comments on the Quality of English LanguagePlease see the uploaded ZIP FILE, which contains 2 Word documents.
Round 2
Reviewer 3 Report
Comments and Suggestions for Authors
(A) The authors have produced an excellent Revised Manuscript.
(B) However, there are a few editorial corrections that need to be made. These are:
----Line 66: Correct to read, "Yet, there has not been....."
---- Line 111: Correct to read, ".....A separate section is dedicated........"
----Line 404: Correct to read, "......Cazenave.....[84]............."
---Line 540: Ref. 45 is not an article by Wallace et al. Correct this error.
----Line 709: Correct to read, "Finding support for this hypothesis....."
----Line 715: Correct to read, "is important to recognize....."
----Line 729: Correct to read,"....may be summarized in six key points..."
----List of References: Correct Cazenave to be Ref. 84 and Volpano to be Ref. #85.
---- Ref. 91 is not cited in the text. Please check this.
Author Response
(A) The authors have produced an excellent Revised Manuscript.
Thank you very much for your kind words. I appreciate the time and effort you have put into this review.
(B) However, there are a few editorial corrections that need to be made. These are:
----Line 66: Correct to read, "Yet, there has not been....."
Thank you. Corrected. Done
---- Line 111: Correct to read, ".....A separate section is dedicated........"
Thank you. Corrected. Done
----Line 404: Correct to read, "......Cazenave.....[84]............."
Thank you. Corrected. Done
---Line 540: Ref. 45 is not an article by Wallace et al. Correct this error.
Thank you. Corrected. Done (lines 527-8)
----Line 709: Correct to read, "Finding support for this hypothesis....."
Thank you. Corrected. Done (line 697)
----Line 715: Correct to read, "is important to recognize....."
Thank you. Corrected. Done (line 703)
----Line 729: Correct to read,"....may be summarized in six key points..."
Thank you. Corrected. Done (line 717)
----List of References: Correct Cazenave to be Ref. 84 and Volpano to be Ref. #85.
Thank you. Corrected. Done
---- Ref. 91 is not cited in the text. Please check this.
Thank you. I think it went unnoticed. See line 601: “"... enhance its efficiency of neural transmissions [91]".